# Key Aroma Differences in Volatile Compounds of Aged Feng-Flavored Baijiu Determined Using Sensory Descriptive Analysis and GC×GC–TOFMS

**DOI:** 10.3390/foods13101504

**Published:** 2024-05-13

**Authors:** Jinmei Ren, Zhijian Li, Wei Jia

**Affiliations:** 1College of Bioresources Chemical & Materials Engineering, Shaanxi University of Science and Technology, Xi’an 710021, China; 2Shanxi Xifeng Liquor Co., Ltd., Baoji 721406, China; 3School of Food Science and Engineering, Shaanxi University of Science and Technology, Xi’an 710021, China; jiawei@sust.edu.cn

**Keywords:** feng-flavored baijiu, sensory descriptive analysis, GC×GC–TOFMS, chemometrics, key differential compounds

## Abstract

Sensory descriptive analysis of aged feng-flavored Baijiu liquor indicated notable differences in samples of different ages. The samples of freshly distilled Baijiu and those with shorter storage times exhibit bran and fresh green flavors, whereas, with increasing storage time, honey, sweet, and floral fragrances are gradually enhanced. Samples of feng-flavored Baijiu were prepared using headspace solid-phase microextraction, followed by comprehensive two-dimensional gas chromatography time-of-flight mass spectrometry. A total of 496 compounds were identified in all samples, mainly categorized as 14 groups of substances, including esters and aldehydes. Interestingly, 42 of these substances were found in Feng-flavored Baijiu for the first time. Chemometrics was used to analyze the key differential compounds. First, 143 differential compounds closely related to aging were preliminarily screened, and principal component analysis revealed that these compounds were separated by baijiu age. Then, 65 differential compounds were selected by partial least squares discriminant analysis. Furthermore, 43 key differential compounds were selected by combined analysis with variable importance in projection and Pearson correlation coefficients. Partial least squares regression was used to study the correlation between the sensory properties and key differential compounds, and the results indicated that most compounds were closely related to the aging period of the Baijiu. The results of this study provide a theoretical basis and reference for flavor research on feng-flavored Baijiu.

## 1. Introduction

Baijiu is a unique liquor distilled in China and is listed among the six foremost distilled liquors globally, including brandy, whisky, rum, vodka, and gin. There are 12 flavor types of Chinese Baijiu, which are classified based on the brewing technique and style characteristics, including feng-flavored and strong-flavored Baijiu. Baijiu is brewed with high-quality sorghum as the raw material and medium-high-temperature Daqu (with a core temperature of 58–60 °C). The raw materials for Daqu of Feng flavored Baijiu are barley and peas, which are fermented for thirty days in a house designed for placing Daqu specifically. The newly fermented Daqu needs to be stored for more than 3 months before they are used for brewing Baijiu. The mixture of sorghum, rice husk and Daqu is finally fermented in an earthen cellar. The fermentation cycle lasts one year, with cellar erection in September and cellar picking in June of the following year. The process involves cellar erection, cellar breaking, top cells, round cells, cellar insertion (May of the following year), and cellar picking. Freshly distilled feng-flavored Baijiu is characterized by a pungent and stimulating taste, which becomes soft and mellow after storage [1]. Baijiu is typically stored in traditional Mare Nectaris containers, which are fabricated using vitex negundo vines (Appendix A), hemp paper, and animal blood, with the addition of egg white, beeswax, and edible oil [2]. Generally, Mare Nectaris containers weigh 5 tons (Appendix A). After pouring freshly distilled Baijiu into Mare Nectaris containers, it undergoes slow physical and chemical reactions (redox, condensation hydrolysis, Maillard reactions, etc.) [3,4,5] and acquires a typical pleasant aroma during long-term storage.

Baijiu contains thousands of volatile compounds in different proportions, resulting in different styles. Although many physical and chemical reactions occur during aging, the chemical reactions mainly determine the sensory quality of Baijiu [6]. To date, the application of various analysis techniques, such as GC-MS (Gas chromatography-mass spectrometry) and GC-MS (Gas chromatography-ion migration spectrometer), has provided a strong background for studying Baijiu; however, these techniques have some drawbacks, such as insufficient peak capacity. Very little research has been published on feng-flavored Baijiu of different ages. Liu et al. adopted GC-MS and GC-IMS to analyze feng-flavored Baijiu and showed differences between Baijiu of different ages; a law of change of the volatile compounds had been derived preliminary [7]. Owing to the resolution limitations of GC–MS and GC–IMS, some trace volatile compounds cannot be detected, and their impact on feng-flavored Baijiu may be overlooked. However, some crucial substances that are previously undetected, unknown, or below the detection limit of GC-MS but greatly influence the style characteristics of Baijiu may be neglected due to the limited separation ability of GC-MS [8]. GC×GC–TOFMS (two-dimensional gas chromatography-time-of-flight mass spectrometry) is an advanced chromatographic separation analytical method that combines two techniques with different separation mechanisms. Nonpolar and long chromatography columns are used in one-dimensional (1D) GC, which separates volatile compounds based on their differences in boiling points. Short chromatography columns of high or moderate polarity are used in two-dimensional (2D) GC to further separate compounds based on their polarity differences [9]. Compared to traditional 1D GC, 2D GC has advantages such as a higher scanning speed, larger peak capacity, faster analysis speed, larger amount of collected information, higher resolution, and shorter analysis time [10].

GC×GC–TOFMS has strong qualitative capabilities and has become a powerful tool for analyzing volatile compounds in recent years [11,12,13]. Headspace solid-phase Microextraction (HS-SPME) is an extensive and effective method for the pretreatment of Baijiu, and many researchers have identified a variety of volatile substances in the sauce, light, strong, fuhe, and other flavors of Baijiu by HS-SPME–GC×GC–TOFMS [14,15,16]. These analysis methods have greatly expanded our understanding of the volatile flavor compounds of Baijiu. Ren et al. [17] preliminary explored volatile flavor compounds in feng-flavored Baijiu by GC×GC–TOFMS. However, owing to the small number of samples, the differential compounds in aged Baijiu have not been deeply investigated.

Chemometrics is an essential means of metabolomic analysis that can deeply mine large datasets and is used to analyze complex systems [18,19,20]. To date, the number of samples analyzed in previous studies was small, which ranged from 10 to 18. The dimensions of the analysis can be reduced through principal component analysis (PCA) and Partial Least Squares Discriminant Analysis (PLS-DA), and the relationship between the data can be intuitively displayed in low dimensions [21]. Jia et al. screened 47 compounds with significant differences in aged Baijiu using ultrahigh-pressure liquid chromatography (UHPLC)-Q-Orbitrap [22]. It was proposed that the interaction between feng-flavored Baijiu and the Mare Nectaris forms a honey fragrance. The storage process is one of the main factors affecting the style characteristics of the liquor.

Few studies have been performed on the characteristic compounds in feng-flavored Baijiu and their relationship with flavor profiles. The evaluation of aged Baijiu is mainly based on sensory evaluations. This study aimed to explore the differences in sensory and volatile compounds of different aged baijiu samples by both sensory evaluation and instrumental analysis and evaluate the correlations between key differential compounds and sensory properties. We expect the results to provide a theoretical reference for flavor characteristic research on feng-flavored Baijiu.

## 2. Materials and Methods

### 2.1. Samples and Reagents

Thirty types of feng-flavored Baijiu of different ages were produced by Shaanxi Xifeng Liquor Co., Ltd., as a representative enterprise of feng-flavored Baijiu. Considering the batch differences in samples of the same age, three samples were selected from different Mare Nectaris of the same age. The samples were stored in a 4 °C refrigerator before detection. In order to facilitate the display of some graphics, the samples of A1, A2, and A3 are represented by A; the values of A were the average of A1 + A2 + A3, similar to others. The sample information is shown in Table 1.

Typical alkanes from C7 to C30 (≥99.8%) were purchased from Sigma (St. Louis, MO, USA). Analytical-grade sodium chloride was purchased from China National Medicines Pharmaceutical Group Corporation (Shanghai, China), and chromatography-grade n-hexyl-d13 alcohol (≥98.5%) was purchased from C/D/N Isotopes Inc. (Quebec, QC, Canada). Chromatography-grade anhydrous ethanol (≥99.8%) was purchased from Aladdin (Shanghai, China). Ultrapure water was prepared using a Milli-Q ultrapure water machine purchased from Millipore (Bedford, MA, USA). 

### 2.2. Volatile Compound Analysis

#### 2.2.1. Preparation of Internal Standard Solution

An appropriate amount of n-hexyl-d13 alcohol was transferred to a volumetric flask, dissolved in 50% (*v*/*v*) ethanol solution to a final concentration of 10 mg/L, and stored in a 4 °C refrigerator.

#### 2.2.2. HS-SPME

An appropriate sample was transferred to a 20 mL glass test tube, diluted with saturated sodium chloride aqueous solution to 10% ethanol concentration (*v*/*v*) [23]. Then, 5 mL of each diluted sample was accurately transferred to a headspace vial. Subsequently, 10 μL of the ISTD solution was added to each sample, and the preprocessed samples were incubated at 50 °C for 10 min. The samples were extracted using a headspace solid-phase microextracter coated with a DVB/CAR/PDMS fiber head (50/30 μm × 1 cm, Supelco, Bellefonte, PA, USA) and kept at 60 °C for 30 min in an incubator. The extracted samples were desorbed in a GC injection port at 250 °C for 5 min and analyzed using GC×GC–TOFMS according to the set parameters. The samples above were subjected to three parallel tests to ensure reproducible experimental results.

#### 2.2.3. GC×GC–TOFMS Method

GC×GC conditions: LECO Pegasus BT 4D (LECO, St. Joseph, MI, USA), 1D column: DB-Heavy Wax (30 m × 250 μm × 0.5 μm) (Agilent, CA, USA), 2D column: Rxi-5Sil MS (2 m × 150 μm × 0.15 μm) (Restek, PA, USA).The initial temperature of the injection port was maintained at 40 °C for 2 min, then increased from 40 to 250 °C at 5 °C/min and maintained at 250 °C for 7 min. High-purity helium (99.99%) was used as the carrier gas with a 1.0 mL/min flow rate. Non-split injection was performed. For 2D analysis, a modulation period of 4.0 s and a thermal pulse period of 1.2 s were used. The temperature of the column was kept at 5 °C hotter than the 1D column.

TOFMS conditions: 70 eV electron bombardment source; ion source temperature of 250 °C; transmission line temperature of 240 °C; and detector voltage of 1984 V. The MS scanning range was *m*/*z* 35–550, and the collection rate was 200 spectra/s.

### 2.3. Descriptive Sensory Analysis

Using the Analytical Methods for Baijiu standard [24] as a reference, a sensory panel composed of 15 professional assessors was established, including 5 national judges and 10 provincial judges (aged 24 to 45). These assessors all have a keen sense of smell, are members of the baijiu evaluation committee of Shaanxi Xifeng Liquor Co., Ltd., Baoji, China, and have extensive experience in sensory evaluation (Appendix A).

The samples were described based on the flavor wheel description of feng-flavored Baijiu [25]. The assessors recorded the aroma and taste intensity and then rated the order from 0 (not perceivable) to 5 (strong) on a scale with increments of 0.5. The samples were evaluated in multiple rounds, with 5 cups each round and a pouring volume of 15–20 mL per cup. The samples were divided into a total of 18 rounds. In order to ensure the sensitivity of the taster’s sense of smell and taste, we choose to evaluate the samples every other day, with 4 rounds per day and 2 rounds on the last day from 9–12 am; the time for each round was up to 30 min, with an interval of 15 min between each round. All samples were evaluated three times, and the obtained sensory values were averaged and plotted on a radar map. All panelists evaluated the aroma profiles of feng-flavored Baijiu.

### 2.4. Statistical Analysis

#### 2.4.1. Data Processing

Data were collected using a Pegasus 4D workstation (LECO). Data analysis and processing were performed using the instrument’s built-in ChromaTOF software, automatically identifying peaks with a signal-to-noise ratio over 50 and comparing them with the NIST14 and Wiley 9 mass-spectrometry libraries to generate a peak table automatically. Compounds with halogen and silicon elements were removed, and chromatographic peaks with forward and reverse similarity greater than 700 were screened using MS. Each volatile compound’s retention index (RI) was calculated for C7–C30 n-alkanes and compared with the RI reference values in the NIST online database (https://webbook.nist.gov/ (accessed on 21 September 2022)). Compounds with RI differences of 50 or less were selected. Finally, compounds with a greater than 50% occurrence rate were selected as reliable results [26]. The content of each flavor compound was calculated using the internal standard method, as shown in Equation (1):(1)$$\mathrm{content}\;\mathrm{of}\;\mathrm{volatile}\;\mathrm{compounds}\;(\mu g/\mathrm{kg})=\frac{\mathrm{peak}\;\mathrm{area}\;\mathrm{of}\;\mathrm{compound}\;\times \mathrm{content}\;\mathrm{of}\;\mathrm{internal}\;\mathrm{standard}\;(\mu g/\mathrm{kg})}{\mathrm{peak}\;\mathrm{area}\;\mathrm{of}\;\mathrm{internal}\;\mathrm{standard}}$$

#### 2.4.2. Chemometric Analysis

The GC×GC–TOFMS data for the feng-flavored baijiu samples were mined and analyzed using a chemometric method. Irrelevant and redundant variables were filtered by preprocessing, thereby improving the effectiveness and accuracy of the data. Excessive missing values can pose difficulties for subsequent analysis steps. Variables with more than 20% of the values missing among the samples were filtered according to the 80% rule. The variables were semi-quantitatively analyzed based on internal standards; therefore, the systematic differences generated during the extraction and detection processes were reduced. A small portion of missing values were interpolated by the K-nearest neighbor algorithm based on machine learning to simplify the univariate and multivariate analyses. The imputation value was half the minimum peak area of the same substance for all samples.

Variables with no statistical significance were removed by univariate analysis, and possible important compounds were preliminarily screened. Single-factor analysis of variance (ANOVA) was performed using SPSS software (IBM SPSS Statistics 22.0). The error-detection rate corrected *p*-values to reduce false-positive results, and the variables were preliminarily screened based on *p* < 0.05 and a Pearson correlation coefficient |r| > 0.6. Data selected by one-way ANOVA was used for multivariate statistical analysis after scaling by unit variance. PCA, PLS-DA, and PLSR were performed using SIMCA-P 14.1 software. Cluster heatmaps and radar maps were plotted using the Biodeep online analysis platform (https://www.biodeep.cn/ (accessed on 21 September 2022)).

## 3. Results

### 3.1. Sensory Analysis Results

The flavor of feng-flavored Baijiu of different ages was evaluated based on the sensory evaluation method, giving the resulting sensory-evaluation radar map shown in Figure 1.

The sensory attributes of feng-flavored Baijiu of different ages were significantly different. There was little difference between fresh Baijiu and that stored for one year. However, after three years of storage, the sensory properties changed significantly. The aroma difference of the samples stored for 10–30 years was greater than those stored for 0–5 years. The fresh green, newly produced, and bran flavors gradually weakened with increasing age, whereas the honey, floral, aged, and Mare Nectaris aromas gradually strengthened. The grain, fruit, alcohol, and nutty aromas initially increased and then slightly weakened. Feng-flavored Baijiu stored for 20 years presented the highest overall sensory score.

The flavor profile of Feng-flavored Baijiu of different ages mainly depends on the volatile flavor compounds detected and analyzed comprehensively in this study. In addition, the aroma and taste of each sample did not change synchronously, as indicated by the sensory analysis results. The samples with longer storage times exhibited higher overall aroma scores, whereas the taste scores initially increased and then slightly decreased over time. Therefore, further in-depth research is needed.

### 3.2. Identification of Volatile Compounds

The samples of Feng-flavored Baijiu were detected by HS-SPME–GC×GC–TOFMS. Ninety chromatograms were extracted, and two representative 2D chromatograms are shown in Figure 2.

The number of volatile compounds in Feng-flavored Baijiu of different ages was significantly different, which is 413, 392, 382, 368, 382, 355, 340, 346, 328, and 330 in sequence from sample A to J. The samples with longer ages had more abundant volatile components than those with shorter ages. GC×GC–TOFMS detected a more comprehensive range and higher quantity of volatile flavor compounds than GC-MS. In total, 496 substances were identified, including 154 esters, 33 acids, 67 alcohols, 32 aldehydes, 27 ketones, 41 olefins, 13 sulfur-containing compounds, 9 lactones, 16 nitrogen-containing compounds, 20 furans, 7 phenols, 35 terpenes, 17 aromatics, and 25 other compounds. The quantity of volatile compounds increased from 330 to 413 with increasing storage time. The quantity of volatile compounds in the 30-year-old Baijiu was ≈1.25 times that in fresh Baijiu (Figure 3a). The quantity of esters was highest in the samples of all ages, followed by alcohols, acids, and aldehydes. In addition, the number of compounds in the Baijiu increased during storage in Mare Nectaris containers, similar to the conclusions of other studies. In addition, with increasing storage age, the relative shares of esters, alcohols, ketones, and sulfur-containing compounds decreased, while those of acids, olefins, nitrogen-containing compounds, furans, terpenes, aromatics, aldehydes, and other compounds tended to increase to varying degrees; the lactones and phenolic compounds did not exhibit a significant variation trend (Figure 3b).

Various flavor compounds were detected by GC×GC–TOFMS, where 42 were identified for the first time in Feng-flavored Baijiu, as shown in Table 2.

The compounds identified for the first time mainly included medium-chain esters, phenyl ring esters, aldehydes, pyrazines, furans, terpenes, lactones, and other compounds. They all have various typical flavors; most phenyl cyclic esters present flavors such as honey, sweet, freedom, floral, rose, balsam, nutty, etc. Medium-chain esters, such as isoamyl nonanoate, ethyl tigoate, and cyclohexyl butyrate, mainly present fruity, floral, oily, and waxy aromas. Most aldehyde compounds present flavors such as fatty, green, waxy, and oily. Ketone compounds present flavors such as sweet and floral, while pyrazine and furan compounds mostly present flavors such as nutty, cocoa, roasted, chocolate, almond, coffee, sweet, etc. Most terpene compounds present woody, balsamic, and sweet characteristics; γ-butyrolactone presents flavors such as cream, oil, sweet, fatty, and caramel. It is worth mentioning that indole presents pleasant flavors such as honey, floral, and jasmine at low concentrations and presents unpleasant odors such as fishy, naphthalene, burnt, and fecal at high concentrations. They may be important markers for distinguishing Feng-flavored Baijiu of different ages, and their influence on the Feng-flavored Baijiu needs to be further studied.

### 3.3. Changes in Volatile Compounds during Storage

Esters are the most diverse substances in Baijiu, which endow it with rich fruit and flower fragrances and prominently contribute to its overall aroma [27]. With increasing storage time, the total ester content significantly decreased. The content of short linear esters (C4–C12), such as ethyl acetate and hexyl hexanoate, decreased. The content of medium linear esters (C12–C15) first increased and then started to drop, while the content of long linear esters (C16–C20) such as myristic acid ethyl ester, palmitic acid ethyl ester, ethyl pentadecanoate, ethyl octadecanoate, and ethyl linoleate, initially increased and then decreased after 20 years. The contents of some branched esters, such as isoamyl nonanoate, isoamyl acetate, and trans-4-decanoate ethyl ester, did not exhibit an obvious pattern but showed an upward trend; ethyl 3-methylbutyrate, ethyl 4-methylvalerate showed a decreasing trend. Changes in the content and type of ester are one of the main reasons for the sensory changes in feng-flavored Baijiu over time.

Acids are important substances that influence the flavor of Baijiu and can blend the taste and stabilize the aroma. If the acid content of Baijiu is too low, its taste is weakened. Sweat and other unpleasant odors are present if its acid content is too high [28,29]. The content of most acids showed an upward trend over time due to the hydrolysis of esters during storage [30]. Owing to the limitations of the GC×GC–TOFMS methods, most acids were not detected in this study. The presence of heptanoic anhydride and octanoic anhydride showed a linearly increasing trend over time, which may be related to the hydrolysis of esters during the aging of Baijiu.

Alcohols are the primary source of mellow sweetness and aroma and the precursor of esters in Baijiu [31]. The boiling points of alcohols are low, making them easy to volatilize, which enriches the flavor of Baijiu [32]. Research has shown that higher alcohols (such as n-propanol, isobutanol, and sec-butanol with >3 carbon atoms) in appropriate concentrations make Baijiu soft and mellow [27]. However, if their concentration is too high, they can impart unpleasant odors to the Baijiu and cause dizziness. The contents and proportions of various higher alcohols are closely related to a pleasant sensation after drinking. The main higher alcohols in feng-flavored Baijiu are n-propanol, sec-butanol, isoamyl alcohol, n-butanol, and isobutanol, which accounted for approximately 85% of the total alcohol content. The contents of isoamyl alcohol (365.45–1087.47 mg/L) and isobutanol (131.78–187.38 mg/L) increased, while the contents of n-propanol (452.31–234.6 mg/L), n-butanol (578.89–153.65 mg/L), 2-nonanol, 2-pentanol, 2-heptanol, 1-octen-3-ol, etc. decreased during storage. Some studies have demonstrated that isopentyl alcohol is a crucial substance correlated with the almond flavor of feng-flavored Baijiu [33], and the ratio of n-propanol to isopentyl alcohol is the main factor determining whether dizziness occurs after drinking. This study indicated that the ratio of n-propanol to isoamyl alcohol in feng-flavored Baijiu decreased from 1.24 to 0.22 during storage, which may be one of the main reasons for the comfort experienced after drinking.

Aldehydes and ketones are important aromatic components of Baijiu, enhancing the aroma and taste of Baijiu [34]. The main aldehydes in feng-flavored Baijiu are acetaldehyde (125.31–54.13 mg/L) and acetal (89.23–210.23 mg/L). Changes in the contents of these two substances improve aging and softness. With increasing age, the contents of benzaldehyde, phenylacetaldehyde, cinnamaldehyde, and saffron volatiles increased. It is worth mentioning that multiple aldehyde compounds were identified in samples stored for a long time. The contents of 2-octenal, 2-heptanenal, and 2-nonenal, which possess green and vegetable aromas [35], showed a decreasing trend, while the content of dienal aldehydes substances (in particular, (E,E)-2,4-nedenal, 2,4-hexadienal, 2,4-decadienal, and 2-methylpentane-2-enal, which impart oil and meat flavors) showed an increasing trend during storage. Ketones, with a lower threshold value and greater aroma contribution, are mainly formed in the fatty acid beta-oxidation process and endow Baijiu with floral, fruity, and creamy flavors [36]. As the age increases, the change in the content of vanillin acetone, isophorone, β-damadone, and acetophenone does not show a linear growth trend. Terpene compounds are important aromatic and physiologically active substances in Baijiu, with antibacterial, antioxidant, and other physiological functions [37,38]. A wide variety of terpenes exist in feng-flavored Baijiu. For example, β-caryophyllene, (+)-longifolene, and (+)-cedarol have sweet and woody aromas, and their contents changed significantly during storage [35]. In addition, cedarol, linalool, etc., were only identified in the samples stored for over 10 years. These compounds may be markers of feng-flavored Baijiu of different ages, and further research is needed to confirm this.

Most lactone compounds provide sweet, fruity, and other aromas, and the flavor of Baijiu becomes softer after the interaction of lactone with other volatile components [39]. The main lactone compounds detected in feng-flavored Baijiu are γ-nonalactone, γ-butyrolactone, and γ-decyl lactone. Apart from γ-nonalactone, no other lactones were detected in fresh Baijiu, which may form slowly in the later storage stages.

Furan and pyrazine compounds mainly impart a pleasant burnt and roasted aroma to Baijiu, where pyrazine compounds are a class of functional substances beneficial to human health [40]. In this study, most furan and pyrazine compounds (including 2,3,5-trimethylpyrazine benzofuran and 2-acetylfuran) were detected in the samples that were stored for more than 10 years, and their contents increased with increasing storage time.

The content of aromatic compounds in Baijiu is minimal (below 0.2%) [41]; however, the threshold value is lower, and the retention time of the odor is longer, which helps improve the quality of Baijiu [42]. Most of the aromatic ester compounds in Baijiu provide fruity, flowery, rose, and honey flavors, among others [43]. In addition to phenylethyl acetate, the contents of ethyl 3-phenylpropionate, ethyl phenylacetate, phenylethyl butyrate, phenylethyl isobutyrate, ethyl benzoate, and isobutyl benzoate increased significantly during storage and are closely related to floral, honey, and fruity aromas. The p-cresol content decreased with increasing storage time; p-cresol was related to the taste of cellar mud [44]; it is inferred that the presence of p-cresol is one of the reasons why fresh Baijiu has an unpleasant and muddy odor, which is consistent with the sensory evaluation results.

### 3.4. Difference Analysis of Volatile Compounds in Feng-Flavored Baijiu

Missing values were filtered and deleted from all samples, and 298 compounds were obtained for the univariate analysis. The results of the one-way ANOVA indicated that 223 compounds changed significantly during storage. Pearson correlation analysis indicated that there was a total of 39 unrelated or weakly correlated compounds (|r| = 0–0.4), 42 moderately correlated compounds (|r| = 0.4–0.6), 101 strongly correlated compounds (|r| = 0.6–0.8), and 41 highly correlated compounds (|r| = 0.8–1) [45].

A total of 143 differential compounds were preliminarily screened, which simultaneously met the three conditions of less than 20% missing values, Pearson correlation coefficient |r| > 0.6, and *p* < 0.05, including 45 esters, 11 alcohols, 15 acids, 19 aldehydes, 9 ketones, 7 furans, 3 phenols, 3 lactones, 4 sulfur-containing compounds, 13 aromatics, 5 terpenes, 5 nitrogen-containing compounds, and 4 other compounds.

Generally, multivariate analysis in chemometrics is divided into unsupervised and supervised techniques. The former is commonly used for preliminary data exploration, including PCA and cluster analysis [46]. Therefore, 143 important compounds selected by single-factor analysis were used for PCA to explore their differences, as shown in Figure 4.

The PCA results showed cumulative contribution rates of PC1 and PC2 of 80.30%, which describe the essential characteristics of the samples and reflect the overall differences. The samples of different ages were separated without significant overlap, indicating that the samples of different ages could be well distinguished. The samples were divided into five groups with certain distributions. The Ji and Ii samples are located in the fourth quadrant and distributed quite close together, whereas the Hi, Gi, and Fi samples are in the first quadrant. The Ei, Di, and Ci samples are clustered in the second quadrant, and the distances between Ei and Di are small, indicating their differences. The Ai and Bi samples, which were stored for more than 20 years, are located in the third quadrant. Greater distances between sample groups indicate greater aroma differences, consistent with the sensory evaluation results.

PLS-DA is a supervised modeling approach; it is usually used to screen variables that contribute significantly to sample classification by establishing a relationship model between omics data and sample categories [47]. To screen key compounds with significant differences in Feng-flavored Baijiu of different ages, a PLS-DA model was established with 143 of the most important compounds, as determined by preliminary screening, and the results are shown in Figure 5. The samples of different ages were separated on the score plot, indicating that the model of PLS-DA is appropriate. In addition, parameters of R2X = 0.911, R2Y = 0.934, and Q2 = 0.8570 were obtained, indicating a good predictive ability (Figure 5a). To verify the reliability of the model, 200 permutation tests were performed to determine whether the model was overfitted. The permutation test gave R2Y = 0.122, Q2 = −0.261, and Q2 < 0.05, indicating that the model did not overfit (Figure 5b).

### 3.5. Identification of Key Compound Differences in Feng-Flavored Baijiu of Different Ages

The contribution of each variable for different sample groups was quantified by the variable importance factor (VIP) of PLS-DA and used to screen important characteristic compounds. Generally, the VIP value is greater than 1, and higher values indicate a better ability to distinguish sample groups. Among the compounds, 65 have VIP values greater than 1; these values and the corresponding r are shown in Table 3.

The 65 potential differential substances mentioned above were screened under VIP > 1 and |r| > 0.7 to screen for the most representative compounds. Finally, 43 compounds were selected, including sixteen (Ethyl acetate, Methyl heptanate, Ethyl hexanoate, Ethyl dodecanoate, Diethyl succinate, Ethyl phenylacetate, 3-Phenylpropanoate ethyl ester, Phenylethyl isobutyrate, Ethyl cinnamate, (2Z)-Butane-2-enoic acid ethyl ester, Butyl isovalerate, Ethyl isovalerate, Ethyl linoleate, Ethyl salicylate, 9-Hexadecaenoic acid ethyl ester, (Z)-Ethyl pentadecene-9-enoate), one acid (γ-linolenic acid); two alcohols (isoamyl alcohol and phenylethanol); seven aldehydes (benzaldehyde, phenylacetaldehyde, furfural, 2-ethyl-2-hexenal, 2-butyl-2-octenal, trans-2-methyl-2-butenal, and 1,1-diethoxyhexane); two ketones (geranylacetone and 3-decanone); three terpenes ((+)-longifolene α- fluorene and cedarol); one lactone (γ-nonalactone); two pyrazine compounds (2,3,5,6-tetramethylpyrazine, 2,3,5-trimethylpyrazine); two sulfur-containing compounds (dimethyl disulfide and dimethyl trisulfide); one phenolic compound (p-cresol); and six other compounds (linaloyl formate, 2-acetylfuran, benzyl ether, α-terpene alcohol, indole, and 2-methylpyridine). Significant contributions to the sensory differentiation of the aged Feng-flavored Baijiu may also be an age marker. Cluster heat maps of the 43 key differential compounds mentioned above are shown in Figure 6. In the clustering heatmap, the samples are well clustered according to the ages and can be divided into four categories, namely, Ji and Ii, Gi and Hi, Di, Ei and Fi, Ai, Bi and Ci. Samples of the same age were clustered together, showing that the 43 compounds effectively distinguish Baijiu of different ages.

### 3.6. Correlation between Sensory Evaluation and Key Differential Compounds

A PLSR model was established based on the 43 key differential compounds in samples of different ages to analyze further the relationship between the sensory properties and key differential compounds. The X variable in the model represents the key differential compounds, the Y variable represents flavor sensory variables, and the Z variable represents samples, as shown in Figure 7.

From the inside out, the three ellipses in Figure 7 represent the explained variance of 50%, 75%, and 100%, respectively. Most key differential compounds and sensory properties lie between 50% and 100% of the explanatory variance, indicating that the model has good interpretability. Significant differences exist in the types of compounds and sensory properties among feng-flavored Baijiu of different ages. Most of the compounds are highly related to the aged Baijiu. The freshly distilled Baijiu (Ji) and 1-year-old Baijiu (Ii) with newly produced flavor, bran flavor, and fresh green flavor are well- and positively correlated mainly with the presence of dimethyl disulfide, dimethyl trisulfide, furfural, and methyl heptanate. The samples stored for 3 and 5 years (Hi and Gi, respectively) with fruit and grain aromas positively correlate with butyl isovalerate, ethyl acetate, and 3-decanone. The samples stored for 10, 15, and 18 years (Fi, Ei, and Di, respectively) exhibited a good correlation with nutty and aged aromas and are positively correlated with benzaldehyde, isoamyl alcohol, 2,3,5,6-tetramethylpyrazine, 1,1-diethoxyhexane, 2,3,5-trimethylpyrazine, benzyl ether, and ethyl cinnamate. The samples stored for 20, 25, and 30 years (Ci, Bi, and Ai, respectively) correlate well with honey, Mare Nectaris, and floral aromas. Ethyl phenylacetate, ethyl 3-phenylpropionate, phenylethyl isobutyrate, and indole correlate well with honey aroma. Vanillin, acetone, phenylacetaldehyde, ethyl myristate, and linalool formate positively correlate with floral aroma. 2-butyl-2-octenal, ethyl linoleic acid, ethyl palmitate, (+)-longifolene, 2-ethyl-2-hexenal, cedarol, 9-hexadecenoic acid ethyl ester, γ- linoleic acid, (2Z)-buty-2-enoic acid ethyl ester, (Z)-pentadecan-9-enoic acid ethyl ester, and others, have a good correlation with the Mare Nectaris flavor, which can be decomposed into woody, aged, oily, and waxy flavors, and may be related to the materials contained in Mare Nectaris containers. The (+)-longifolene and cedarol are correlated with woody aromas, whereas 2-ethyl-2-hexenal, trans-2-methyl-2-butenal, and 2-butyl-2-octenal are correlated with oily aromas. In addition, the samples stored for 20, 25, and 30 years are positively correlated with bitter and sweaty compounds such as 2-methylpyridine and p-cresol. This indicates that in addition to fruity and sweet aroma attributes, aged Baijiu has some unpleasant aromas.

## 4. Conclusions

This study applied sensory descriptive analysis to analyze the sensory attributes and volatile substances of feng-flavored Baijiu detected by HS-SPME-GC×GC-TOFMS. The research results indicated notable differences in the aromas of the samples of different ages. Freshly distilled and one-year-old Baijiu both exhibited bran and fresh green aromas. Samples aged 3–5 years had distinct fruity and grain aromas, while those aged 10–20 had mellow and nutty aromas, indicating their older age. The samples aged 20–30 years had significant honey, Mare Nectaris, and floral aromas.

In total, 496 volatile substances were identified by HS-SPME-GC×GC–TOFMS and 42 crucial trace volatile compounds were discovered in feng-flavored Baijiu for the first time. It is preliminarily inferred that the increase in volatile compounds with lower content and stronger aroma may be key factors in improving the quality of Baijiu. Then, 143 compounds closely related to age were preliminarily screened using chemometric methods. The PCA analysis showed that the feng-flavored baijiu samples were well clustered according to the storage period. In addition, 65 differential compounds with VIP > 1 were screened based on PLS-DA. Based on the VIP values and Pearson correlation coefficients, 43 key differential compounds of Feng-flavored Baijiu were ultimately selected, including 18 esters, 1 acid, 2 alcohols, 7 aldehydes, 2 ketones, 3 terpenes, 1 lactone, 2 pyrazines, 2 sulfur-containing compounds, 1 phenol, and 6 other compounds. The results also showed that esters are important substances for distinguishing the aroma of aged Baijiu. The clustering heatmap analysis further confirmed the effectiveness of the 43 key substances in differentiating the aged baijiu samples. A correlation evaluation model between key differential compounds and sensory properties of aged Baijiu was established using PLSR, which indicated that most of the key differential compounds are correlated with the flavor characteristics of the samples with longer storage times.

Based on these results, it is preliminarily inferred that the Mare Nectaris flavor related to terpenes, aldehydes, and long-chain esters, long-chain ethyl ester, and aldehydes may be an important marker for distinguishing feng-flavored Baijiu of different ages. Synergistic or additive effects occur in compounds with similar structures or aromas, which may increase or decrease certain aroma intensities, such as ethyl compounds with benzene rings. However, the mechanism underlying the proportion and composition of various compounds on the quality of Baijiu needs to be further studied.

This study enriches the knowledge related to flavor chemistry and provides a reference for flavor- and sensory-oriented research on feng-flavored Baijiu, which is very important for analyzing its quality.

## Figures and Tables

**Figure 1 foods-13-01504-f001:**
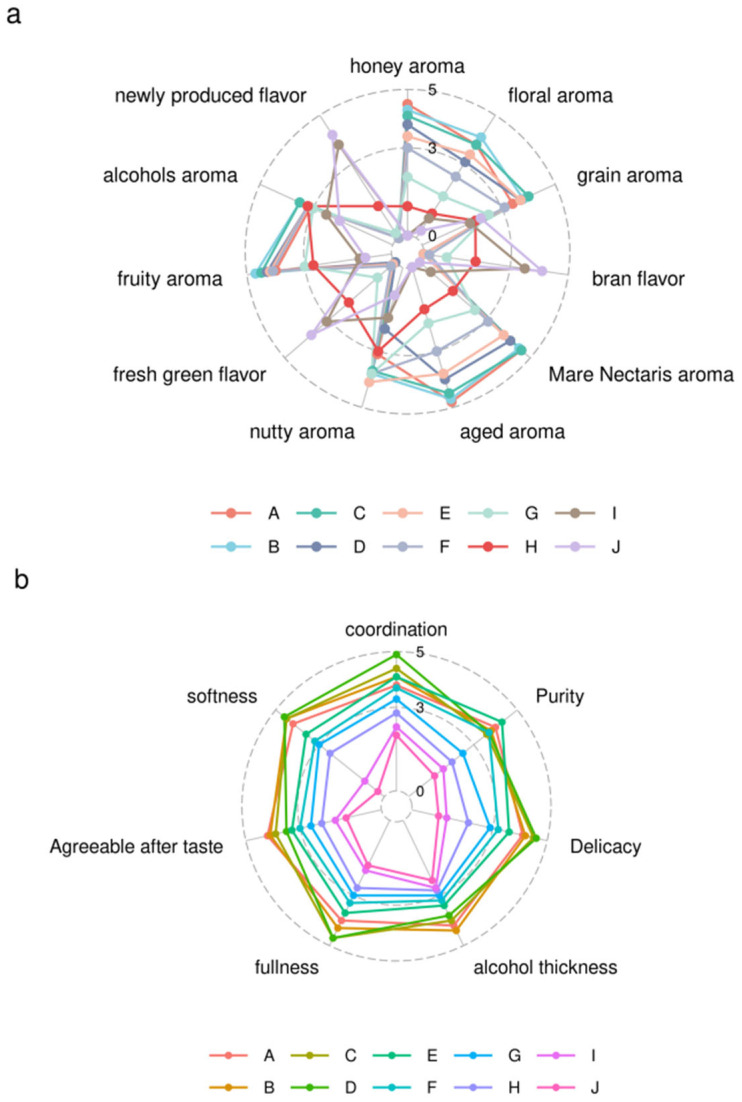
Sensory-evaluation radar map of Feng-flavored Baijiu of different ages. (**a**) Aroma. (**b**) Taste.

**Figure 2 foods-13-01504-f002:**
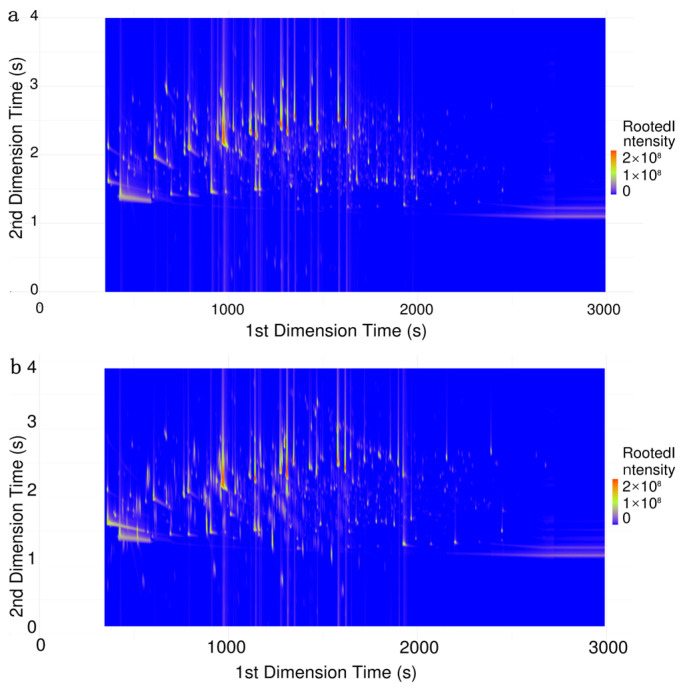
D total chromatography spectra of Feng-flavored Baijiu. (**a**) The sample of A1. (**b**) The sample of J1.

**Figure 3 foods-13-01504-f003:**
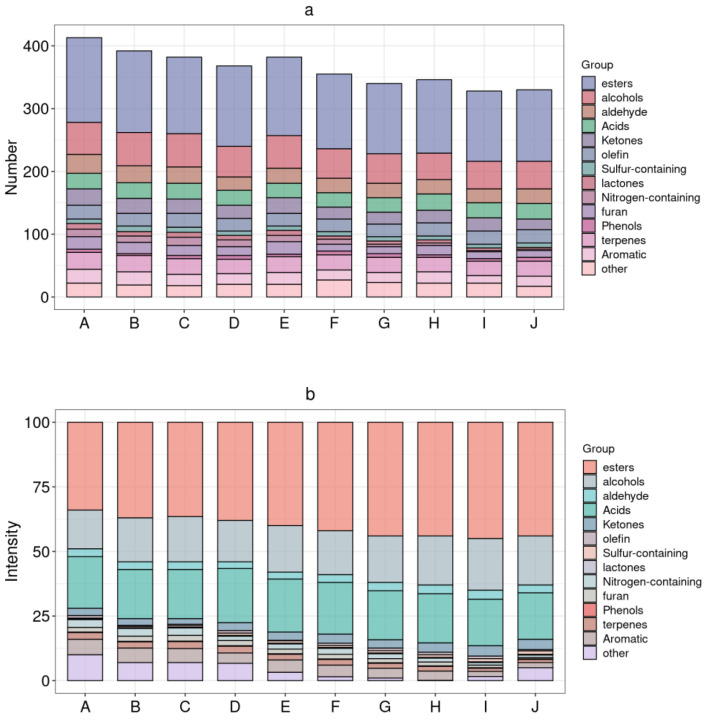
Volatile flavor compounds in Feng-flavored Baijiu. (**a**) Types and quantities of volatile compounds. (**b**) Relative percentage of the volatile compounds.

**Figure 4 foods-13-01504-f004:**
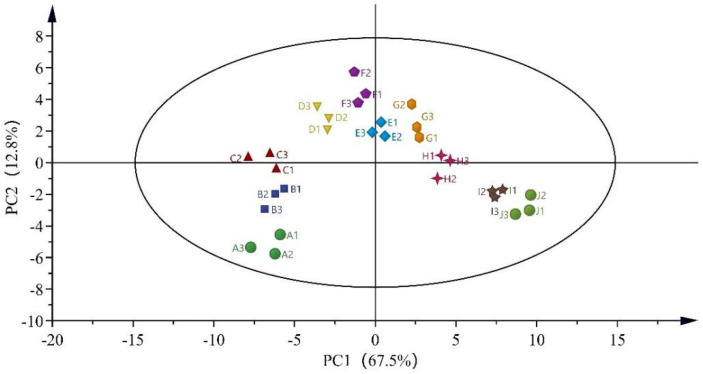
PCA analysis score plots of Feng-flavored Baijiu of different ages.

**Figure 5 foods-13-01504-f005:**
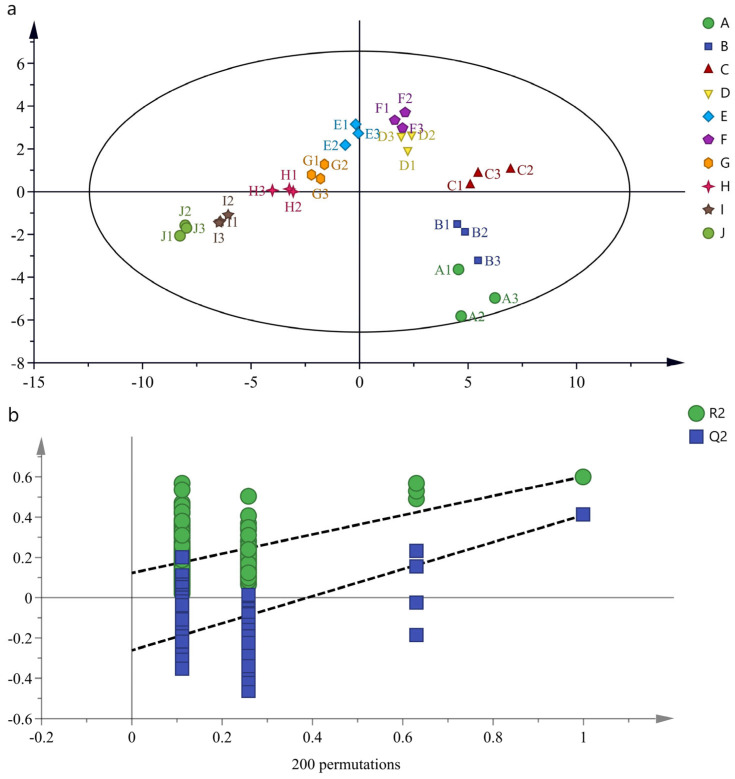
PLS-DA analysis of Feng-flavored Baijiu of different ages. (**a**) Score plot of PLS-DA and (**b**) permutation plot.

**Figure 6 foods-13-01504-f006:**
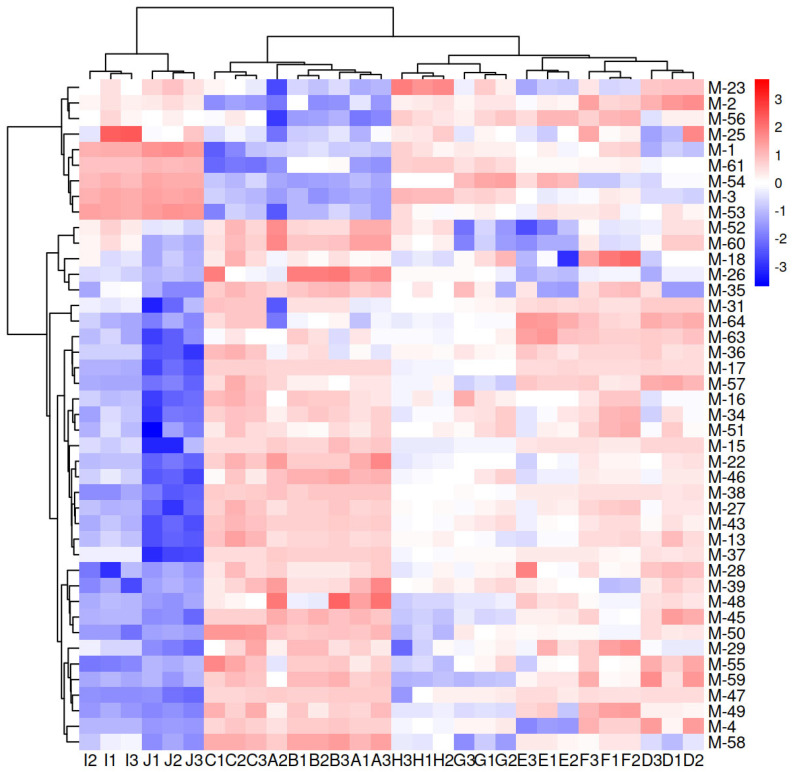
Cluster heatmap of 43 key differential volatile compounds.

**Figure 7 foods-13-01504-f007:**
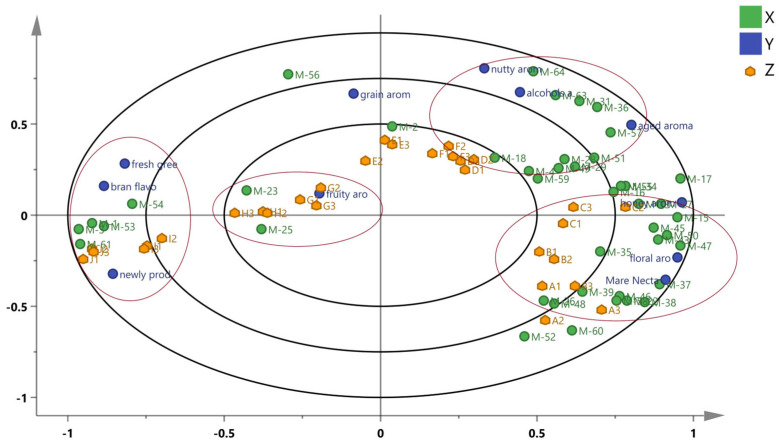
PLSR analysis of key differential volatile compounds and sensory properties.

**Table 1 foods-13-01504-t001:** Information Table the Samples.

Number	Samples	Ages	Number	Samples	Ages
A	A1	30	F	F	10
	A2	30		F	10
	A3	30		F	10
B	B1	25	G	G1	5
	B2	25		G2	5
	B3	25		G3	5
C	C1	20	H	H1	3
	C2	20		H2	3
	C3	20		H	3
D	D1	18	I	I	1
	D2	18		I	1
	D3	18		I	1
E	E1	15	J	J1	0
	E2	15		J2	0
	E3	15		J3	0

**Table 2 foods-13-01504-t002:** Major volatile compounds identified in Feng-flavored Baijiu for the first time.

Compounds	1DRt /s	2DRt/s	Similarity	Relative Content (ug/L)										Oaroma Descriptors
				A	B	C	D	E	F	G	H	I	J	
Isoamyl nonanoate	1789.91	2.68	861	287.45 ± 16.63 ^a^	321.11 ± 55.48 ^a^	166.87 ± 19.22 ^b^	ND	175.12 ± 95.56 ^b^	151.55 ± 75.18 ^b^	143.67 ± 21.19 ^b^	110.51 ± 122.88 ^c^	129.45 ± 19.63 ^c^	129.33 ± 58.13 ^c^	fruity, floral, oily, nutty
Ethyl tiglate	982.59	1.89	945	21.54 ±5.21 ^ab^	21.77 ±1.39 ^ab^	11.75 ± 5.21 ^c^	24.21 ± 0.77 ^a^	11.03 ± 2.3 ^c^	4.65 ± 1.82 ^d^	ND	ND	ND	ND	fruity, raspberry, floral, sweet,
Cyclohexyl butyrate	1271.94	2.41	912	71.91 ± 0.99 ^a^	75.43 ± 3.66 ^a^	65.66 ± 2.84 ^ab^	41.74 ± 0.83 ^c^	52.95 ± 5.21 ^b^	40.79 ± 1.42 ^c^	22.75 ± 2.89 ^d^	16.83 ± 1.73 ^d^	ND	ND	fruity, green, apple, waxy
Propyl benzoate	1801.91	1.71	845	11.51 ± 0.98 ^b^	8.62 ± 1.02 ^bc^	9.34 ± 1.05 ^b^	14.18 ± 1.26 ^a^	17.59 ± 1.36 ^a^	10.40 ± 2.45 ^b^	3.82 ± 0.29 ^c^	2.72 ± 0.31 ^d^	3.08 ± 0.44 ^d^	2.02 ± 0.25 ^d^	balsam, nutty
Butyl benzoate	1941.9	1.76	873	13.02 ± 1.21 ^c^	23.19 ± 1.88 ^b^	27.68 ± 2.56 ^b^	57.79 ± 4.22 ^a^	60.39 ± 3.15 ^a^	20.57 ± 1.99 ^b^	8.00 ± 1.05 ^d^	12.00 ± 1.58 ^c^	8.20 ± 1.81 ^d^	7.26 ± 1.23 ^d^	balsamic, fruity
Benzyl acetate	1761.91	1.53	938	1.52± 0.02 ^d^	2.27 ± 0.03 ^c^	3.44 ± 0.15 ^c^	3.04 ± 0.06 ^c^	1.29 ± 0.07 ^d^	8.40 ± 1.06 ^b^	4.43 ± 0.57 ^bc^	20.79 ± 5.38 ^a^	2.27 ± 0.09 ^c^	2.54 ± 0.17 ^c^	boiled vegetable, fruity, jasmine, fresh
Butyl phenylacetate	2050.54	1.73	880	8.15 ± 0,21 ^a^	9.76 ± 0.77 ^a^	9.64 ± 0.73 ^a^	5.80 ± 0.24 ^b^	7.01 ± 1.15 ^ab^	5.60 ± 0.19 ^b^	5.57 ± 0.27 ^b^	3.22 ± 0.16 ^c^	2.12 ± 0.03 ^d^	1.39 ± 0.18 ^d^	honey, rose
Isobutyl phenylacetate	1973.9	1.74	845	6.22 ± 0.33 ^ab^	6.68 ± 1.02 ^a^	7.26 ± 0.87 ^a^	5.57 ± 0.43 ^b^	5.07 ± 0.17 ^b^	4.67 ± 0.24 ^c^	3.92 ± 0.32 ^c^	4.29 ± 0.18 ^c^	3.01 ± 0.36 ^d^	2.56 ± 0.15 ^d^	honey, floral, sweet
Phenylethyl propionate	1789.91	1.7	934	5.93 ± 0.49 ^a^	5.77 ± 1.03 ^a^	6.51 ± 0.885 ^a^	4.82 ± 0.54 ^b^	3.76 ± 0.49 ^bc^	1.04 ± 0.07 ^d^	2.51 ± 0.12 ^c^	0.30 ± 0.01 ^d^	ND	ND	rose, fruity, raspberry and strawberry
Phenylethyl isobutyrate	1919.02	1.7	867	676.91 ± 45.89 ^a^	636.21 ± 78.89 ^ab^	749.63 ± 36.58 ^a^	631.41 ± 49.38 ^ab^	615.01 ± 59.29 ^ab^	416.85 ± 29.41 ^b^	591.80 ± 66.37 ^b^	152.88 ± 10.02 ^c^	64.31 ± 5.39 ^d^	74.62 ± 3.68 ^d^	fruity, sweet, rose
Isobutyl benzoate	1840.2	1.78	839	67.09 ± 2.67 ^a^	54.13 ± 2.57 ^ab^	ND	44.12 ± 2.46 ^b^	31.50 ± 2.45 ^c^	2.20 ± 0.23 ^d^	ND	2.69 ± 0.16 ^d^	ND	ND	sweet, fruity, floral aroma similar to roses and fragrant leaves
2-Heptanenal	1131.04	1.78	945	ND	ND	1.00 ± 0.06 ^d^	1.56 ± 0.18 ^c^	1.84 ± 0.21 ^c^	2.84 ± 0.05 ^bc^	3.50 ± 0.21 ^b^	4.62 ± 0.15 ^b^	7.95 ± 0.74 ^b^	91.18 ± 4.47 ^a^	fatty, green, fresh, pungent
2-nonenal	1385.2	2.09	835	21.46 ± 2.23 ^d^	41.24 ± 2.16 ^cd^	52.77 ± 3.18 ^c^	74.65 ± 5.23 ^b^	70.33 ± 5.38 ^b^	91.93 ± 6.47 ^bc^	157.72 ± 13.73 ^b^	206.35 ± 19.25 ^b^	402.18 ± 27.65 ^a^	357.23 ± 29.95 ^ab^	green, melon, fatty, cucumber, waxy
2-Octenal	1293.69	1.91	866	1.77 ± 0.08 ^d^	2.66 ± 0.04 ^d^	19.15 ± 1.56 ^cd^	32.84 ± 2.65 ^c^	39.53 ± 2.96 ^c^	51.11 ± 4.74 ^bc^	58.53 ± 3.88 ^b^	64.53 ± 5.99 ^b^	87.98 ± 4.87 ^a^	66.47 ± 4.72 ^b^	herbal, leaf, green, fresh, fatty, waxy
2,4-Hexadienal	1243.63	1.52	954	30.09 ± 2.35 ^a^	35.79 ± 2.44 ^a^	30.77 ± 1.98 ^a^	26.82 ± 1.77 ^b^	21.12 ± 2.46 ^b^	18.84 ± 1.34 ^bc^	12.21 ± 1.27 ^c^	10.35 ± 1.02 ^c^	5.38 ± 0.48 ^d^	3.13 ± 0.21 ^d^	green, floral, sweet, citrus, spicy
2,4-Decadienal	1865.9	1.71	831	85.78 ± 5.67 ^ab^	101.62 ± 7.38 ^a^	94.91 ± 4,29 ^a^	87.76 ± 5.16 ^ab^	67.97 ± 3.28 ^b^	56.86 ± 2.19 ^b^	39.50 ± 2.17 ^bc^	28.76± 2.01 ^c^	3.03 ± 0.06 ^d^	1.95 ± 0.15 ^d^	fatty, waxy, green, oily, meat, fresh
2-ethyl-2-hexenal	1149.95	1.88	853	377.50 ± 23.31 ^bc^	594.35 ± 45.29 ^a^	476.86 ± 55.39 ^b^	340.21 ± 23.17 ^bc^	253.34 ± 15.98 ^c^	212.23 ± 24.76 ^c^	113.65 ± 8.16 c^d^	90.22 ± 4.29 ^d^	38.19± 3.27 ^d^	2.18 ± 0.05 ^e^	fatty, waxy, green, oily,
2-Butyl-2-octenal	1662.95	2.1	877	399.81 ± 15.92 ^b^	422.11 ± 55.21 ^b^	577.08 ± 24.37 ^a^	368.42 ± 28.49 ^b^	251.88 ± 36.29 ^c^	192.23 ± 13.15 ^c^	102.40 ± 5.29 ^d^	72.18 ± 4.89 ^d^	38.47 ± 3.32 ^de^	2.18 ± 0.53 ^e^	fatty, waxy, green, oily,
β-Cyclocitral	1613.17	1.87	919	5.18 ± 0.78 ^d^	7.01 ± 1.21 ^bc^	10.78 ± 1.02 ^a^	10.74 ± 1.77 ^a^	9.92 ± 0.96 ^a^	9.12 ± 0. 25 ^ab^	8.14 ± 0.29 ^b^	7.99 ± 1.01 ^b^	6.47 ± 0.16 ^c^	4.89 ± 0.24 ^d^	minty, fruity, herbal, sweet, tobacco
ZangHonghua-aldehyde	1641.55	1.75	855	36.00 ± 2.45 ^b^	51.06 ± 4.19 ^a^	43.39 ± 1.87 ^a^	28.68 ± 3.45 ^b^	15.08 ± 1.19 ^c^	10.31 ± 1.04 ^c^	5.05 ± 0.53 ^d^	4.62 ± 1.01 ^d^	ND	ND	woody, medicina, Spicy
Isophorone	1446.06	1.73	877	4.30 ± 0.66 ^b^	4.57 ± 0.23 ^b^	ND	ND	5.45 ± 1.34 ^ab^	2.88 ± 0.27 ^c^	6.80 ± 0.54 ^a^	5.91 ± 0.23 ^a^	4.18 ± 0.19 ^b^	1.54 ± 0.06 ^d^	fruity, sweet, cedarwood, camphoraceous, musty, wood
Acetophenone	1668.53	1.51	911	90.34 ± 5.59 ^b^	91.59 ± 6.43 ^b^	125.44 ± 10.92 ^a^	88.56 ± 4.32 ^b^	76.25 ± 1.27 ^c^	119.28 ± 7.99 ^a^	88.23 ± 2.18 ^b^	37.09 ± 1.21 ^cd^	24.35 ± 1.65 ^d^	10.76 ± 0.83 ^d^	sweet, floral, pungent, almond, chemical
β-Ionone	2009.42	1.03	863	1.18 ± 0.03 ^a^	0.82 ± 0.05 ^b^	0.48 ± 0.11 ^c^	0.97 ± 0.09 ^b^	0.56 ± 0.06 ^c^	0.28 ± 0.03 ^d^	ND	ND	ND	ND	woody, raspberry, violet, floral
Anisole	1166.33	1.62	956	2.17 ± 0.18 ^a^	2.19 ± 0.23 ^a^	1.88 ± 0.14 ^b^	2.34 ± 0.27 ^a^	1.08 ± 0.36 ^c^	1.34 ± 0.19 ^bc^	0.57 ± 0.03 ^d^	0.32 ± 0.11 ^d^	ND	ND	anise, Aromatic, phenolic,
Dagenxiangye	1396.41	0.33	892	69.48 ± 2.91 ^c^	186.74 ± 16.97 ^a^	162.08 ± 12.56 ^a^	104.03 ± 4.56 ^b^	79.93 ± 5.12 ^c^	79.70 ± 3.73 ^c^	53.10 ± 2.56 ^d^	35.04 ± 1.67 ^d^	ND	ND	/
2,6-Dimethylpyrazine	1129.95	1.64	914	5.05 ± 0.87 ^a^	1.87 ± 0.12 ^c^	2.54 ± 0.25 ^b^	1.74 ± 0.13 ^c^	ND	1.65 ± 0.06 ^c^	1.99 ± 0.32 ^bc^	1.19 ± 0.21 ^d^	ND	ND	roasted nut, cocoa, coffee
2,3,5-Trimethylpyrazin-e	1257.94	1.72	925	151.36 ± 13.12 ^a^	147.81 ± 9.08 ^a^	117.81 ± 13.98 ^b^	104.32 ± 5.54 ^b^	83.27 ± 7.19 ^bc^	61.71 ± 3.22 ^c^	30.50 ± 3.45 ^c^	14.62 ± 1.11 ^d^	3.79 ± 1.09 ^d^	2.06 ± 0.88 ^d^	nutty, cocoa, musty, peanut, earthy, potato
2,3,5,6-TetramethylPyr-azine	1369.93	1.8	887	89.47 ± 14.23 ^a^	68.26 ± 8.67 ^b^	62.50 ± 5.44 b^c^	43.95 ± 3.12 ^b^	42.91 ± 3.02 ^b^	36.75 ± 4.09 ^c^	24.90 ± 3.25 ^c^	20.55 ± 3.72 ^c^	9.62 ± 1.29 ^d^	4.23 ± 0.58 ^d^	burnt, soybean, chocolate, cocoa, nutty, coffee, musty, lard
2-Acetofuran	1433.93	1.41	846	33.77 ± 5.43 ^a^	22.66 ± 3.26 ^b^	26.36 ± 4.12 ^b^	19.20 ± 1.46 ^bc^	8.86 ± 2.87 ^d^	14.02 ± 3.76 ^c^	16.80 ± 3.49 ^c^	14.69 ± 2.11 ^c^	4.59 ± 1.59 ^d^	5.29 ± 2.12 ^d^	balsamic, cocoa, almond, sweet, tobacco, peanut, caramel, potato
2-Acetyl-5 Methylfuran	1624.82	1.49	881	42.81 ± 1.47 ^c^	85.15 ± 5.67 ^a^	91.85 ± 6.34 ^a^	77.68 ± 4.37 ^b^	78.88 ± 5.31 ^b^	60.35 ± 4.61 ^bc^	55.82 ± 3.27 ^bc^	44.15 ± 5.46 ^c^	27.99 ± 2.49 ^d^	25.99 ± 3.64 ^d^	nutty, coconut, milky, musty, hay
α-Guyunene	1477.05	3.39	957	2.08 ± 0.24 ^d^	3.14 ± 0.55 ^d^	2.70 ± 0.46 ^d^	1.13 ± 0.03 ^d^	9.18 ± 1.11 ^c^	14.66 ± 2.23 ^b^	17.30 ± 3.37 ^a^	15.41 ± 2.98 ^ab^	17.23 ± 3.15 ^a^	19.47 ± 1.25 ^a^	balsamic, balsam, woody
Cedarol	2261.88	1.97	829	8.51 ± 0.65 ^b^	10.39 ± 1.23 ^b^	7.34 ± 1.35 ^b^	14.95 ± 3.32 ^a^	3.60 ± 0.97 ^c^	2.53 ± 0.88 ^c^	ND	3.42 ± 0.48 ^c^	1.83 ± 0.07 ^d^	1.50 ± 0.05 ^d^	cedarwood, sweet, soft, woody
Aromatic ene	997.23	2.29	947	ND	ND	3.87 ± 0.44 ^d^	3.31 ± 0.11 ^d^	12.13 ± 2.31 ^c^	16.81 ± 1.64 ^c^	35.78 ± 2.46 ^b^	ND	54.31 ± 9.37 ^a^	45.82 ± 10.33 ^a^	woody
Nerol	1897.9	1.54	912	15.30 ± 2.59 ^d^	14.70 ± 2.76 ^b^	17.90 ± 3.47 ^b^	12.60 ± 1.98 ^bc^	10.30 ± 1.76 ^bc^	22.50 ± 3.45 ^a^	7.76 ± 1.31 ^c^	7.54 ± 1.52 ^c^	4.36 ± 0.88 ^d^	4.39 ± 0.29 ^d^	sweet, citrus, neroli
β-Caryophyllene	1615.47	2.99	939	17.03 ± 2.25 ^e^	64.75 ± 9.56 ^d^	82.11 ± 8.89 ^d^	83.26 ± 10.54 ^d^	100.46 ± 12.78 ^c^	108.09 ± 13.46 ^c^	126.22 ± 11.77 ^b^	146.22 ± 23.37 ^b^	199.72 ± 45.79 ^ab^	225.85 ± 56.34 ^a^	spice, sweet, clove, woody
Longifolene	1547.81	3.16	892	36.24 ± 3.24 ^b^	66.11 ± 5.56 ^a^	44.77 ± 4.66 ^b^	15.21 ± 1.99 ^c^	13.99 ± 2.35 ^c^	11.28 ± 1.21 ^c^	5.05 ± 0.68 ^d^	6.62 ± 1.22 ^d^	2.04 ± 0.14 ^de^	0.50 ± 0.09 ^e^	rose, sweet, medical, woody
indole	2605.86	1.28	873	3.41 ± 0.33 ^ab^	4.10 ± 0.75 ^a^	3.99 ± 0.15 ^a^	3.04 ± 0.25 ^b^	2.44 ± 0.41 ^c^	1.82 ± 0.12 ^d^	1.47 ± 0.31 ^d^	1.65 ± 0.18 ^d^	ND	ND	honey, floral, jasmine, fishy, naphthalene, burnt, fecal
Linaloyl formate	1477.93	1.67	855	185.95 ± 83.26 ^a^	233.88 ± 74.16 ^a^	129.82 ± 54.39 ^b^	105.61 ± 43.27 ^b^	85.15 ± 21.89 ^bc^	68.22 ± 10.46 ^c^	40.06 ± 10.85 ^d^	24.62 ± 2.25 ^cd^	5.51± 1.29 ^d^	5.24 ± 1.61 ^d^	herbal, fruity, rose, citrus
γ-Butyrolactone	1706.51	1.38	887	6.85 ± 0.55 ^a^	2.69 ± 0.31 ^b^	1.69 ± 0.18 ^c^	1.69 ± 0.23 ^c^	1.89 ± 0.19 ^c^	1.68 ± 0.61 ^c^	1.33 ± 0.15 ^c^	0.84 ± 0.22 ^d^	ND	ND	creamy, oily, sweet, fatty, caramel
Dimethyl disulfide	669.98	1.71	875	ND	ND	4.11 ± 0.29 ^d^	7.71 ± 1.63 ^d^	18.21 ± 5.45 ^c^	24.01 ± 3.99 ^c^	80.98 ± 15.72 ^b^	103.83 ± 46.77 ^b^	173.52 ± 77.34 ^a^	187.00 ± 56.76 ^a^	cabbage, green, putrid, nutty, sulfurous, onion
P-cresol	2197.88	1.25	934	13.23 ± 2.25 ^de^	6.02 ± 1.56 ^e^	16.12 ± 3.12 ^de^	22.77 ± 2.23 ^d^	51.23 ± 9.96 ^c^	85.82 ± 15.46 ^bc^	101.22 ± 23.49 ^b^	128.45 ± 32.88 ^b^	155.44 ± 25.47 ^a^	172.38 ± 44.31 ^a^	smoke, phenol, phenolic, medicine, narcissus, mimosa, animal, medicinal
2-Methylpyridine	922.12	1.63	937	111.20 ± 15.69 ^a^	84.43 ± 12.35 ^b^	66.45 ± 9.44 ^bc^	28.45 ± 6.52 ^c^	9.35 ± 0.13 ^cd^	2.43 ± 0.55 ^d^	ND	5.18 ± 0.95 ^d^	ND	ND	bitter, sweaty, unpleasant

ND: not detected. /: aroma descriptors were not identified. Different letters (a, b, c, d, e) indicate significant differences between different samples.

**Table 3 foods-13-01504-t003:** Potential differential volatile compounds in feng-flavored Baijiu of different ages.

Number	Compound	CAS	VIP	r	Number	Compound	CAS	VIP	r
M-1	Ethyl acetate	141-78-6	1.49	−0702	M-34	Phenylacetaldehyde	122-78-1	1.82	0.808
M-2	Methyl heptanate	106-73-0	2.07	−0.718	M-35	Phenylethanol	1960-12-8	1.13	0.707
M-3	Ethyl hexanoate	123-66-0	1.21	−0.711	M-36	Isoamyl alcohol	123-51-3	1.55	−0.715
M-4	Ethyl dodecanoate	106-33-2	1.53	0.809	M-37	2-ethyl-2-hexenal	645-62-5	2.11	0.832
M-5	Ethyl myristate	124-06-1	1.45	0.721	M-38	2-Butyl-2-octenal	13019-16-4	1.87	0.855
M-6	2-Methylbutyrate ethyl ester	7452-79-1	1.35	0.631	M-39	Trans-2-methyl-2-butenal	497-03-0	2.01	0.804
M-7	Butyl butyrate	109-21-7	1.51	−0.605	M-40	2-Methylpent-2-enal	623-36-9	1.06	0.675
M-8	Hexyl Hexanoate	6378-65-0	1.09	−0.627	M-41	Octanoic anhydride	623-66-5	1.21	0.654
M-9	Diethyl azelaic acid	624-17-9	1.51	0.754	M-42	Heptanic anhydride	626-27-7	1.67	0.628
M-10	Ethyl octanoate	106-32-1	1.62	−0.601	M-43	3-Decanone	693-54-9	1.02	0.816
M-11	Isobutyl heptanate	7779-80-8	1.32	0.625	M-44	Damascus ketone	23726-93-4	1.12	0.687
M-12	Trans-4-decanoate ethyl ester	76649-16-6	1.67	0.681	M-45	Geranyl acetone	3796-70-1	1.42	0.822
M-13	Diethyl succinate	123-25-1	1.43	0.755	M-46	α-Terpenoid alcohol	98-55-5	1.25	0.763
M-14	3-Ethyl furoate	614-98-2	1.76	0.692	M-47	Linaloyl formate	115-99-1	1.06	0.757
M-15	Ethyl phenylacetate	101-97-3	1.21	0.833	X-48	Cedarol	77-53-2	1.52	0.813
M-16	3-Phenylpropanoate ethyl ester	2021-28-5	1.65	0.844	M-49	Benzyl ether	100-66-3	1.12	0.677
M-17	Phenylethyl isobutyrate	103-48-0	1.71	0.805	M-50	γ-Nonlactone	104-61-0	1.14	0.734
M-18	Ethyl cinnamate	103-36-6	1.43	0.831	M-51	Ethyl palmitate	628-97-7	1.22	0.824
M-19	2,4-Hexadienal	142-83-6	1.84	0.783	M-52	2-Methylpyridine	109-06-8	1.68	0.761
M-20	3-hydroxy-2-butanone	513-86-0	1.09	0.688	M-53	Dimethyl disulfide	624-92-0	1.65	−0.746
M-21	2-Ethyl Nonenoate	17463-01-3	1.03	0.637	M-54	Dimethyl trisulfide	3658-80-8	1.85	−0.813
M-22	(2Z)-Butane-2-enoic acid ethyl ester	6776-19-8	1.07	0.803	M-55	Indole	120-72-9	1.89	0.871
M-23	Butyl isovalerate	109-19-3	1.93	0.826	M-56	γ-linolenic acid	506-26-3	1.75	0.803
M-24	2-Ethyl heptaneoate	54340-72-6	1.4	0.659	M-57	1,1-diethoxyhexane	3658-93-3	1.63	0.603
M-25	Ethyl isovalerate	108-64-5	1.16	−0.712	M-58	α- Fluorene	3856-25-5	1.49	0.745
M-26	Ethyl linoleate	2089036	1.03	0.814	M-59	2-Acetofuran	1192-62-7	1.06	0.805
M-27	Ethyl salicylate	118-61-6	1.42	0.809	M-60	(+)-Long leafene	475-20-7	1.35	0.722
M-28	9-Hexadecaenoic acid ethyl ester	54546-22-4	1.52	0.824	M-61	Furfural	35796	1.31	0.712
M-29	(Z)-Ethyl pentadecene-9-enoate	56219-09-1	1.85	0.811	M-62	Benzofuran	271-89-6	1.17	0.658
M-30	Isobutanol	78-83-1	1.09	0.618	M-63	2,3,5-Trimethylpyrazine	14667-55-1	1.29	0.776
M-31	Benzaldehyde	100-52-7	1.54	0.722	M-64	2,3,5,6-Tetramethylpyrazine	1124-11-4	1.42	0.814
M-32	Isovaleraldehyde	590-86-3	1.63	0.682	M-65	Paracresol	106-44-5	1.56	−0.809
M-33	2-nonenal	18829-56-6	1.69	−0.664					

## Data Availability

The original contributions presented in the study are included in the article/Appendix A, further inquiries can be directed to the corresponding author.

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
