# Peer review of "Key Aroma Differences in Volatile Compounds of Aged Feng-Flavored Baijiu Determined Using Sensory Descriptive Analysis and GC×GC–TOFMS"

_foods, 2024, doi:10.3390/foods13101504_

Round 1
Reviewer 1 Report
Comments and Suggestions for Authors
M;anuscript foods-2981207
The manuscript uses QDA to evaluate the aged Feng-flavored baijiu. With this objective, the samples were analysed by headspace solid-phase microextraction combined with comñrehensive two-dimensional gas chromatography time-of-flight mass spectrometry. The results were statistically analysed using diverse chemometric techniques (PCA,PLS-DA, PLSR, etc.) to relate sensory characteristics with volatile compounds.
Specific comments
General.
Results and Results and Discussion sections are not clearly distinguishable since the first also includes many comments on the results. Furthermore, the second, also repeat ideas already expressed. It is suggested to pass those comments in the second section to the first when appropriate, suppress the current Results and Discussion section and introduce a new one devoted to conclusions.
Revise the text. There are several spaces and other presentation aspects to revise. The list below is not exhaustive.
L34 daqu?
L44. Revise line
L63-64. Revise text
L76. Complete reference in text. Also, in other paragraphs throughout the text.
L90. Revise
L129-130. Separation is rare
L149. Analyzing 5 cups in each round sounds a bit excessive. Please justify. Were some measures taken to prevent saturation?
L166. Revise
L182. Please add details of the program (version, trade mark, company identification, etc)
Figure 1. Please identify the different samples in the legend of Figure 1 to facilitate interpretation. Also, increase the size of the Figure to allow reading the legends.
L197. Revise verb tense tense
L204-206. The sentence expresses an obvious link. Consider its elimination.
Fig 2. The legends in the graph are difficult to read. Please increase their size. In addition, add information on the legends to facilitate figure understanding.
L217. Was the significance identified just by appreciation or a result of a statistical test?
Fig 3. The meaning of A,B,C,…. This should be explained in the figure legend. By the way, how was constructed the Figure 3b?
Table 1. Explain the meaning of A, B, …- Additionally, explain the meaning of the values in the table. Also, what is the meaning of, e.g. 287.45±16.63. If the second value indicated the variability, it should be more convenient to put it in parenthesis since the use of ± is justified just in the case of confidence limits.
L279. Maybe ~ is not the best sign to indicate an interval due to its association with mathematical functions (LGM, etc., models widely used in R programming).
Section 3.3. Many compounds are related to different aromas, flavors, etc. Please support such associations.
L335-337. How was established the scale?
Fig 4. Please provide information in the legend regarding the meaning of the symbols in the graph.
L352. How was estimate this significance?
Fig 5. Is split. Please combine it as an Image before its inclusion in the text. This could prevent the paragraph from separating the two parts of the graph. Additionally, also include an explanation of symbols in the legend to facilitate understanding.
Fig 6. Add the meaning of samples and indicate where the symbols for compounds can be found.
L404. The heatmap only shows clusters for volatile.
Fig 7. Also, add information, at least where the association of symbols for volatile and their respective compounds can be found. Explain the presence of t(corr)[2] in the symbols’ box.
Comments on the Quality of English LanguageRequire revision.
Author Response
We appreciate for the honorable editors and reviewers for your valuable feedback. As per suggestions of reviewers, we have made substantial modifications to our manuscript. In the revised version, changes to our manuscript were marked red and blue(There is no difference, it's just that the colors are different because they were modified twice). Please see below, in blue and red, for a point-by-point response to the reviewers’ comments and concerns.
Thank you again for your positive comments and valuable suggestions to improve the quality of our manuscript.
Responds to the reviewers’ comments:
Comment 1. Line 34; daqu?
Reply: Thank you for your question. It is our negligence that we failed to describe clearly, and we have revised it. Daqu is saccharifying starter for Baijiu brewing.
As lines 34: “daqu” has been changed to “Daqu”As lines 34-36: added“The raw materials for Daqu of Feng flavored Baijiu are barley and peas, which are fermented for thirty days in a house designed for placing Daqu specifically.The newly fermented Daqu needs to be stored for more than 3 months before they are used for brewing Baijiu”.
Comment 2. Line 44; Revise line.
Reply: Thanks very much for your reminder and the line has been corrected in the revised manuscript to maintain consistency with the template.
As lines 43.
Comment 3. Line 63-64; Revise word .
Reply: Thank you for your reminder. It is our negligence that we failed to separate the two words and we have revised it.
As lines 66: “techniqueswith has been changed to techniques with”.
Comment 4. Line 76; Complete reference in text. Also, in other paragraphs throughout the text.
Reply: Thank you very much for your suggestion, we have revised it..
As lines 80: Ren et al.[16] preliminary explored volatile flavor compounds in Feng-flavored baijiu by GC×GC–TOFMS.
Comment 5. Line 90; Revise.
Reply: Thank you very much for your suggestion, we have revised it.
As lines 94-96: Few studies have been performed on the characteristic compounds in feng-flavored Baijiu and their relationship with flavor profiles. The evaluation of aged Baijiu is mainly based on sensory evaluations.
Comment 6. Line 129-130; Separation is rare.
Reply: Thank you very much for your reminder, we have revised it.
As lines 136-137: 1D column: TR FFAP (30 m × 0.25 mm, 0.25 μm;Thermo Fisher Scientific).
Comment 7. Line 149; Analyzing 5 cups in each round sounds a bit excessive. Please justify. Were some measures taken to prevent saturation?
Reply: Thank you for your valuable advice. I will provide a detailed description of sensory evaluation in this study. The 5-cup method in each round is a standard tasting method for Baijiu evaluation in China and also the most common methods for participants in this experiment to evaluate liquor. If there are too few samples in each round, such as 1-2, errors can easily occur between the evaluation results of each round.
As lines 157-159: added” The samples were divided into a total of 18 rounds. In order to ensure the sensitivity of the taster's sense of smell and taste, we choose to evaluate the samples every other day, with 4 rounds per day and 2 rounds on the last day from 9-12 am, the time for each round was up to 30 min, with an interval of 15 min between each round”.
Comment 8. Line 166; Revise
Reply: Thank you very much for your reminder , It is our negligence that we failed to
align the two lines of paragraphs and we have revised it.
As lines 176-177: The content of each flavor compound was calculated using the internal standard method, as shown in equation (1):
Comment 9. Line 182; Revise. Please add details of the program (version, trade mark, company identification, etc)
Reply: Thanks very much for your suggestion, We have added detailed information about the program.
As lines 193: Single-factor analysis of variance (ANOVA) was performed using SPSS software (IBM SPSS Statistics 22.0). The error-detection rate corrected P-values to reduce false-positive results.
Comment 10. Figure 1. Please identify the different samples in the legend of Figure 1 to facilitate interpretation. Also, increase the size of the Figure to allow reading the legends.
Reply: Thank you very much for your reminder. It is our negligence that we failed to describe clearly, we have revised it.
As lines 220: Figure 1.
Comment 11. Line 197; Revise verb tense
Reply: Thank you very much for your reminder , we have revised it.
As lines 207: However, after three years of storage, the sensory properties changed significantly.
Comment 12. Line 204-206; The sentence expresses an obvious link. Consider its elimination.
Reply: Thank you very much for your valuable advice. The sentence have been revised and the taste radar map of Feng-flavored Baijiu of different ages have been added.
As lines 214-217: The flavor profile of Feng-flavored Baijiu of different ages mainly depends on the volatile flavor compounds detected and analyzed comprehensively in this study. In addition, the aroma and taste of each sample did not change synchronously, as indicated by the sensory analysis results.
Comment 13. Fig 2.The legends in the graph are difficult to read. Please increase their size. In addition, add information on the legends to facilitate figure understanding.
Reply: Thank you very much for your reminder , It is our negligence that we failed to
annotate the legend to a reasonable size and we have revised it.
As lines 229: Figure 2.
Comment 14. Line 217. Was the significance identified just by appreciation or a result of a statistical test?
Reply: Thank you very much for your question. The significance identified by appreciation.
As lines 233-234: added”which is 413, 392, 382, 368, 382, 355, 340, 346, 328, and 330 in sequence from sample A to J”.
Comment 15. Fig 3. The meaning of A,B,C,…. This should be explained in the figure legend. By the way, how was constructed the Figure 3b?
Reply: Thank you very much for your reminder and valuable suggestion.
Due to the large number of samples in the experiment, it is not suitable to display all of them. In addition, A1, A2, and A3 are parallel samples from the same year. Therefore, we use A to represent the samples of A1, A2, and A3, the values of A was the average of A1+A2+A3. similarly, the B,C,…..J.
Regarding the construct of Figure 3b: The data was calculated using the following method. The proportion of the total relative content of all compounds in each sample is considered to be 100%, The specific calculation formula is as follows: X=Ci*100%/C
X:The percentage content of a certain type of compound in the sample;
Ci:The total relative content of a certain type of compound in the sample;
C:The total relative content of all compounds in the sample.
As lines 110: The meaning of A,B,C,…. is shown in Table 1.
Table 1. Information Table of the Samples.
|
Number |
Samples |
Ages |
Number |
Samples |
Ages |
|
A |
A1 |
30 |
F |
F |
10 |
|
A2 |
30 |
F |
10 |
||
|
A3 |
30 |
F |
10 |
||
|
B |
B1 |
25 |
G |
G1 |
5 |
|
B2 |
25 |
G2 |
5 |
||
|
B3 |
25 |
G3 |
5 |
||
|
C |
C1 |
20 |
H
|
H1 |
3 |
|
C2 |
20 |
H2 |
3 |
||
|
C3 |
20 |
H |
3 |
||
|
D |
D1 |
18 |
I |
I |
1 |
|
D2 |
18 |
I |
1 |
||
|
D3 |
18 |
I |
1 |
||
|
E |
E1 |
15 |
J |
J1 |
0 |
|
E2 |
15 |
J2 |
0 |
||
|
E3 |
15 |
J3 |
0 |
Comment 16. Table 1. Explain the meaning of A, B, …Additionally, explain the meaning of the values in the table. Also, what is the meaning of, e.g. 287.45±16.63. If the second value indicated the variability, it should be more convenient to put it in parenthesis since the use of ± is justified just in the case of confidence limits.
Reply:Thanks very much for the valuable advice, we fully agree with your opinions. The meaning of A, B,... is shown in Table 1. the compare statistical analysis was added in revised manuscript,significance markers has been added in the table 2.
As lines 273 : table 2.
Comment 17. Line 279. Maybe ~ is not the best sign to indicate an interval due to its association with mathematical functions (LGM, etc., models widely used in R programming)
Reply: Thanks very much for the valuable advice, we fully agree with your opinions, “~ ”has been changed to“–”
As lines 308-310: The contents of isoamyl alcohol (365.45–1087.47 mg/L) and isobutanol (131.78–187.38 mg/L) increased, while the contents of n-propanol (452.31–234.6 mg/L), n-butanol (578.89–153.65 mg/L)........
Comment 18. Section 3.3. Many compounds are related to different aromas, flavors, etc. Please support such associations.
Reply: Thanks very much for the valuable advice.
Regarding the associations between compounds and aromas, flavors , we referred to relevant literature and professional books(which has been listed). In addition,we conducted sensory evaluations on standard solutions of some compounds in the laboratory, such as ethyl acetate, ethyl hexanoate, ethyl lactate, and ethyl butyrate, but not entirely including the compounds we mentioned. Further research is needed to confirm the correlation between these compounds and their aroma.Which compounds play a leading role in a certain aroma of Baijiu is also a necessary research for my topic to continue. Thank you very much for your guidance.
As lines 633-635, 654-657: added the following references
- Wang L L, Gao M X, Liu Z P, Chen S, Xu Y. Three extraction methods in combination with GC x GC-TOFMS for the detailed investigation of volatiles in chinese herbaceous aroma-type Baijiu. Molecules, 2020, 25(19): 4429. DOI:3390/molecules25194429
- FangC, Liu Z. G, Qiao L, Huang J.L. Analysis of volatile characteristic flavors of three aroma types of Shanzhuang Laojiu by sensory quantitative descriptive analysis and gas chromatography-mass spectrometry[J]. Food Sci. 2023, 44(10), 291-299. DOI:10.7506/spkx1002-6630-20220821-241.
- Jia, Z. Y. Chinese Baijiu tasting treasure, 1st ed,; Chemical Industry Press: Beijing, China, 2016; pp. 22-36.
Comment 19. Line 335-337; How was established the scale?
Reply: Thanks very much for the question, the scale was established by referencing algorithms literat”Abnormal Frame Detection in Elevator IoT Data Collection Based on Pearson-LOF Algorithm”. It is also an important method for screening differential compounds from a large number of samples at present.
Comment 20.Fig 4. Please provide information in the legend regarding the meaning of the symbols in the graph.
Reply: Thanks very much for the reminder. It is our negligence that we failed to describe clearly, the meaning of the symbols(A1,A2,A3.....JI,J2,J3) is shown in table1.
Comment 21. Line 352; How was estimate this significance?
Reply: Thanks very much for your question. The “significance” was estimated by observation previously, each symbol represents a sample,the symbols of all samples do not overlap in the legend.
Comment 22. Fig 5. Is split. Please combine it as an Image before its inclusion in the text. This could prevent the paragraph from separating the two parts of the graph. Additionally, also include an explanation of symbols in the legend to facilitate understanding.
Reply: Thank you very much for your reminder and valuable suggestion. We have combine them as an Image. Additionally, the symbols in the legend have also been reorganized
As lines 413: Fig 5.
Comment 23. Fig 6. Add the meaning of samples and indicate where the symbols for compounds can be found.
Reply: Thanks very much for the valuable advice. It is our negligence that we failed to describe clearly, the meaning of samples is shown in Table 1(as lines 110 ). In addition, the symbols for compounds can be found in table 3(as lines 422).
Comment 24. Line 404; The heatmap only shows clusters for volatile.
Reply: Thanks very much for the reminder. We have provided detailed clustering heatmap, both the volatile compounds and samples were also clustered in the heatmap.
As lines 443-445: addeed ”In the clustering heatmap, the samples are well clustered according to the ages and can be divided into four categories, namely, Ji and Ii , Gi and Hi, Di, Ei and Fi, Ai, Bi and Ci.”
As lines 447: Fig 6.
Comment 25. Fig7. Also, add information, at least where the association of symbols for volatile and their respective compounds can be found. Explain the presence of t(corr)[2] .
Reply: Thank you very much for your valuable suggestion, we fully agree with your opinions. The closer the distance between the samples, compounds and sensory descriptors, the closer the correlation. We have circled the sensory descriptors and compounds related to the sample. In addition, t(corr)[2] means”samples” in the symbols’ box. For better understanding, we have replaced it with” Z“.
As lines 453-454: added” and the Z variable represents samples, ”

Reviewer 2 Report
Comments and Suggestions for Authors
Review of the article no. foods-2981207
Title: Key Aroma Differences in Volatile Compounds of Aged Feng-flavored Baijiu Determined using Sensory Descriptive Analysis and GC×GC–TOFMS
The paper describes volatile compounds present in Feng-flavored baijiu through analysis performed using GC×GC–TOFMS, and statistical analysis of the obtained research results was carried out. This study aims to investigate the differences in sensory and volatile compounds of aged baijiu through sensory evaluation combined with instrumental tests, further analysis of the correlation between key differential compounds and sensory properties, and to provide a theoretical reference for research on the flavor characteristics of Feng-flavored baijiu. The article requires thorough text editing, including language correction. There are many typos and punctuation errors throughout the text, which makes reading difficult.
Introduction
-
Line 34: What is "gaqu"?
-
Line 40: What is "stimulation" in this context?
-
Line 64: Techniqueswith?
-
Line 82: Please indicate how many samples were analyzed.
Results
-
Table 1: Why was statistical analysis not given in the table? Please add it.
-
Lines 219-220: This information was given in the Materials and Methods section.
-
Lines 231-235: Please change the expression "content" into "share" in the profile.
-
Lines 246-249: Please describe those compounds in detail. It would be very interesting to indicate what kind of aroma notes those compounds are contributing to.
-
Line 301: Please change the word "fluctly."
-
Lines 324-330: Can authors discuss this part with other experimental results?
-
Lines 386-390: In the brackets, only esters should be listed, but other compounds can be found. Please check this.
-
Lines 472-474: Please rewrite this sentence; it is unclear.
Can you please include the Conclusions section?
Comments on the Quality of English LanguageThe article should be corrected.
Author Response
We appreciate for the honorable editors and reviewers for your valuable feedback. As per suggestions of reviewers, we have made substantial modifications to our manuscript. In the revised version, changes to our manuscript were marked red and blue(There is no difference, it's just that the colors are different because they were modified twice). Please see below, in blue and red, for a point-by-point response to the reviewers’ comments and concerns.
Thank you again for your positive comments and valuable suggestions to improve the quality of our manuscript.
Responds to the reviewers’ comments:
Comment 1. Line 34; What is “daqu”?
Reply: Thank you for your question. It is our negligence that we failed to describe clearly, and we have revised it. Daqu is saccharifying starter for Baijiu brewing.
As lines 34: “daqu” has been changed to “Daqu”.
As lines 34-36: added“The raw materials for Daqu of Feng flavored Baijiu are barley and peas, which are fermented for thirty days in a house designed for placing Daqu specifically.The newly fermented Daqu needs to be stored for more than 3 months before they are used for brewing Baijiu”.
Comment 2. Line 40; What is "stimulation" in this context?
Reply: Thank you for your question. “stimulation” means :When tasted Baijiu, the taste is pungent, and there is an uncomfortable feeling in the mouth.
As lines 42-43: Freshly distilled feng-flavored Baijiu is characterized by a pungent and stimulating taste.
Comment 3. Line 64; Techniques with?
Reply: Thank you for your question. It is our negligence that we failed to separate the two words and we have revised it.
As lines 66: “techniqueswith has been replaced by techniques with”.
Comment 4. Line 82: Please indicate how many samples were analyzed.
Reply: Thank you for your reminder. the relevant information has been added in the revised manuscript.
As lines 86:“To date, the number of samples analyzed in previous studies was small, which ranged from 10 to 18.”
Comment 5. Table 1: Why was statistical analysis not given in the table? Please add it.
Reply:Thanks very much for the valuable advice, we fully agree with your opinions the compare statistical analysis was added in revised manuscript. significance markers has been added in the table 2.
As lines 273: table 2.
Comment 6. Line 219-220; This information was given in the Materials and Methods section.
Reply: Thanks very much for your reminder and the relevant information has been removed in the revised manuscript to reduce the repetition of text.
As lines 233-234: The amount of volatile compounds in Feng-flavored Baijiu of different ages was significantly different, , which is 413, 392, 382, 368, 382, 355, 340, 346, 328, and 330 in sequence from sample A to J.
Comment 7. Line 231-235; Please change the expression "content" into "share" in the profile.
Reply: Thank you very much for your valuable suggestion. We fully agree with your opinions. We have changed "contents" into "shares".
As lines 246: In addition, with an increase in storage age, the relative shares of esters, alcohols, ketones, and 231 sulfur-containing compounds decreased.
Comment 8. 246-249; Please describe those compounds in detail. It would be very interesting to indicate what kind of aroma notes those compounds are contributing to.
Reply: Thanks very much for the valuable advice, we have provided a more detailed description of those compounds. The aroma descriptors for the compounds were added in table 2 of the revised manuscript.
As lines 257-271: added “They all have various typical flavors, most phenyl cyclic esters present flavors such as honey, sweet, freedom, floral, rose, balsam, nutty, etc. medium-chain esters such as isoamyl nonanoate, ethyl tigoate, and cyclohexyl butyrate mainly present fruity, floral, oily, and waxy aroma. Most aldehyde compounds present flavors such as fatty, green, waxy, and oily. Ketone compounds present flavors such as sweet and floral, while pyrazine and furan compounds mostly present flavors such as nutty, cocoa, roasted, chocolate, almond, coffee, sweet, etc. Most terpene compounds present woody, balsamic, and sweet characteristics, γ- Butyrolactone presents flavors such as cream, oil, sweet, fatty, and caramel. It is worth mentioning that indole presents pleasant flavors such as honey, floral, and jasmine at low concentrations, and presents unpleasant odors such as fishy, naphtholene, burnt, fecal at high concentrations. They may be important marker for distinguishing Feng-flavored baijiu of different ages.”
Comment 9. Line 301; Please change the word "fluctly"
Reply: Thanks very much for the valuable advice, we have revised it.
As lines 329-331: changed to “ As the increasing of ages ,the change in the content of vanillin acetone, isophorone, β-damadone, and acetophenone does not show the linear growth trend” .
Comment 10. Line 324-330; Can authors discuss this part with other experimental results?
Reply: Thank you very much for your valuable suggestion. Regarding the question you raised, I have made appropriate modifications in the corresponding paragraph of the article.
As lines 351-360: The content of aromatic compounds in Baijiu is minimal (below 0.2%) [39]; however, the threshold value is lower, and the retention time of the odor is longer, which helps improve the quality of Baijiu [40]. Most of the aromatic ester compounds in Baijiu provide fruity, flowery, rose, and honey flavors, among others. In addition to phenylethyl acetate, the contents of ethyl 3-phenylpropionate, ethyl phenylacetate, phenylethyl butyrate, phenylethyl isobutyrate, ethyl benzoate, and isobutyl benzoate increased significantly during storage and are closely related to floral, honey, and fruity aromas. The p-cresol content decreased with increasing storage time, p-cresol was related to the taste of cellar mud; it is inferred that the presence of p-cresol is one of the reasons why fresh Baijiu has an unpleasant and muddy odor, which is consistent with the sensory evaluation results.
Comment 11. Line 386-390; In the brackets, only esters should be listed, but other compounds can be found. Please check this.
Reply: Thanks very much for your reminder and valuable suggestion, It is our negligence that we failed to check this carefully. The relevant information has been corrected in the revised manuscript.
As lines 426-430: Finally, 43 compounds were selected, including sixteen (Ethyl acetate, Methyl heptanate, Ethyl hexanoate, Ethyl dodecanoate, Diethyl succinate, Ethyl phenylacetate, 3-Phenylpropanoate ethyl ester, Phenylethyl isobutyrate, Ethyl cinnamate, (2Z)-Butane-2-enoic acid ethyl ester, Butyl isovalerate,Ethyl isovalerate,Ethyl linoleate, Ethyl salicylate, 9-Hexadecaenoic acid ethyl ester, (Z) - Ethyl pentadecene-9-enoate)
Comment 12. Line 472-474; Please rewrite this sentence; it is unclear.
Reply: Thank you very much for your reminder and suggestion, we have expressed the sentence more clearly and accurately at the end of the conclusion in the revised manuscript.
As lines 521-523: This study enriches the knowledge related to flavor chemistry and provides a reference for flavor- and sensory-oriented research on feng-flavored Baijiu, which is very important for analyzing its quality.
Comment 13. Line 472-474; Can you please include the Conclusions section?
Reply: Thank you very much for your meaningful comments, It is our negligence that we failed to distinguish the concept clearly. We have already revised it.
As lines 488-523:4. Conclusions. This study applied sensory descriptive analysis to analyze the sensory attributes and volatile substances of feng-flavored Baijiu detected by HS-SPME-GC×GC-TOFMS. The research results indicated notable differences in the aromas of the samples of different ages. Freshly distilled and one-year-old Baijiu both exhibited bran and fresh green aromas. Samples aged 3–5 years had distinct fruity and grain aromas, while those aged 10–20 had mellow and nutty aromas, indicating their older age. The samples aged 20–30 years had significant honey, Mare Nectaris, and floral aromas.
In total, 496 volatile substances were identified by HS-SPME-GC×GC–TOFMS, and 42 crucial trace volatile compounds were discovered in feng-flavored Baijiu for the first time. It is preliminarily inferred that the increase in volatile compounds with lower content and stronger aroma may be key factors in improving the quality of Baijiu. Then, 143 compounds closely related to age were preliminarily screened using chemometric methods. The PCA analysis showed that the feng-flavored baijiu samples were well clustered according to the storage period. In addition, 65 differential compounds with VIP>1 were screened based on PLS-DA. Based on the VIP values and Pearson correlation coefficients, 43 key differential compounds of Feng-flavored Baijiu were ultimately selected, including 18 esters, 1 acid, 2 alcohols, 7 aldehydes, 2 ketones, 3 terpenes, 1 lactone, 2 pyrazines, and 2 sulfur-containing compounds, 1 phenol, and 6 other compounds. The results also showed that esters are important substances for distinguishing the aroma of aged Baijiu. The clustering heatmap analysis further confirmed the effectiveness of the 43 key substances in differentiating the aged baijiu samples. A correlation evaluation model between key differential compounds and sensory properties of aged Baijiu was established using PLSR, which indicated that most of the key differential compounds are correlated with the flavor characteristics of the samples with longer storage times.
Based on these results, it is preliminarily inferred that the Mare Nectaris flavor related to terpenes, aldehydes, and long-chain esters, long-chain ethyl ester, and aldehydes may be an important marker for distinguishing feng-flavored Baijiu of different ages. Synergistic or additive effects occur in compounds with similar structures or aromas, which may increase or decrease certain aroma intensities, such as ethyl compounds with benzene rings. However, the mechanism underlying the proportion and composition of various compounds on the quality of Baijiu needs to be further studied.
This study enriches the knowledge related to flavor chemistry and provides a reference for flavor- and sensory-oriented research on feng-flavored Baijiu, which is very important for analyzing its quality.

Round 2
Reviewer 1 Report
Comments and Suggestions for Authors
The authors have addressed most of the suggestions. Nor further comments
Comments on the Quality of English LanguageNo important issues were detected.